# Development of a Set of Indicators for Measuring and Improving Quality of Rehabilitation Care after Ischemic Stroke

**DOI:** 10.3390/healthcare11142065

**Published:** 2023-07-19

**Authors:** Maria Cristina De Cola, Augusto Ielo, Francesco Corallo, Sebastiano Pollina Addario, Salvatore Scondotto, Alessandra Allotta, Giovanna Fantaci, Placido Bramanti, Rosella Ciurleo

**Affiliations:** 1IRCCS Centro Neurolesi Bonino-Pulejo, 98124 Messina, Italy; mariacristina.decola@irccsme.it (M.C.D.C.); francesco.corallo@irccsme.it (F.C.); bramanti.dino@gmail.com (P.B.); rossella.ciurleo@irccsme.it (R.C.); 2Assessorato della Salute, Dipartimento Attività Sanitarie e Osservatorio Epidemiologico, 90145 Palermo, Italy; walter.pollina.ext@regione.sicilia.it (S.P.A.); salvatore.scondotto@gmail.com (S.S.); aleallotta@gmail.com (A.A.); giovanna.fantaci@regione.sicilia.it (G.F.); 3Faculty of Psychology, Università Degli Studi eCampus, Via Isimbardi 10, 22060 Novedrate, Italy

**Keywords:** quality indicators, ischemic stroke, rehabilitation, neurorehabilitation, audit and feedback, care pathways

## Abstract

Stroke is the leading global cause of permanent disability and the second leading cause of dementia within the first year of the event. Systematic quality improvement interventions such as Audit & Feedback (A&F) can monitor and improve the performance of post-stroke care in conjunction with the use of quality indicators (QIs). The scientific literature shows limited studies on quality improvement and QIs design for poststroke rehabilitation. In Italy, the National Outcomes Evaluation Programme (PNE) annually provides several QIs concerning the acute wards. On the contrary, indicators for quality assessment of post-acute stroke rehabilitation are not available nationwide. In recent years, the Italian Ministry of Health has funded a national network project, the aim of which is to provide and evaluate the effectiveness of A&F strategies in healthcare improvement. Part of this project is the development of a set of IQs for ischemic stroke rehabilitation used to conduct an A&F. In this study, we describe the design and development process of these QIs from administrative databases and report the results of the pilot test conducted on a small sample of Sicilian rehabilitation facilities, comparing them from 2019 to 2021. Feedback from the participating centers was mainly positive, and the quality indicators were found to be comprehensible and appreciated. However, the study highlighted the need for better adherence to indicators measuring processes of rehabilitation care. The set of quality indicators presented in this study, relevant to inpatient settings, could be considered a starting point on which to base quality improvement initiatives both nationally and internationally.

## 1. Introduction

Cardiovascular (CVD) and cerebrovascular diseases, including heart diseases (heart attacks) and strokes, are the first cause of morbidity and mortality globally [1,2]. In the European Union, CVDs account for approximately 37.1% of all deaths [3], while stroke is the leading cause of permanent disability and, after Alzheimer’s disease, the second leading cause of dementia [4].

Implementing effective strategies for preventing and treating stroke plays a key role in clinical risk management. In this regard, several national and international clinical guidelines for stroke management have been published in recent years in both the acute and post-acute phases [5,6,7]. Indeed, adherence to post-stroke rehabilitation guidelines has been demonstrated to be strongly linked with improved functional outcomes for patients [8].

Italian guidelines for the prevention and treatment of stroke are coordinated and regularly updated by the Italian Stroke Association (ISA-AII) [9]. In Italy, there were more than 70,000 hospitalizations for ischemic stroke in 2020, with a one-year mortality rate of 17.6% [10,11]. Although the most recent statistics have shown a decline in mortality and disability, the burden of cerebrovascular diseases is still high, and the aging population will have a major impact on the incidence rate of this disease, with an expected increase in the elderly affected by stroke. Furthermore, given the better management of acute patients, the number of stroke survivors is increasing, leading to an increased risk of stroke recurrence and the need for post-stroke rehabilitation facilities [12].

### 1.1. Measuring Stroke Care Performances

Monitoring clinical outcomes is a relevant aspect of assessing the quality of healthcare in terms of access to and effectiveness of care. Constant review of clinical outcomes establishes standards that may be addressed to promote continuous improvement in healthcare practice.

Several studies suggest that introducing public reporting programs on clinical outcomes at different levels could be a rewarding strategy for public health [13,14,15]. Notably, the literature shows that quality indicators are widely used to measure hospital performance in order to provide appropriate stroke care [16,17,18]. Thus, clinical outcomes can be measured by hospital activity data or by surveys, agreed scales, and other forms of measurement. They can be recorded by administrators or by clinical staff such as doctors, nurses, psychologists, or other health professionals (e.g., physiotherapists).

In Italy, several research programs on outcomes have been conducted during the last 20 years. The “Mattoni-Outcome Project” and the subsequent “Progr.Es.Si” programs evaluated outcome and process indicators in different clinical and surgical areas and were the starting points for the National Outcomes Evaluation Program, called PNE [19], the purpose of which is public disclosure and comparison of hospital performance data.

The PNE evaluation is based on the use of nationally available data flows [20]. In particular, the sources are the hospital discharge records database (SDO) for Italian public and private accredited clinical facilities [21], the National information system for monitoring care in Emergency-Urgency (EMUR) [22], and the National Tax Registry for verification of patients living status.

The PNE calculates and publishes every year an increasing number of indicators on the quality of care. In 2021, the indicators were 194, of which 171 related to hospital care and 23 concerned avoidable hospitalization and outpatient outcomes. Several clinical areas are covered by the PNE indicators, including CVDs and stroke. In particular, five stroke quality indicators were published by PNE in 2021, although only for acute stroke care monitoring: (i) number of ischemic strokes, (ii) 30-day mortality, (iii) one-year mortality, (iv) 30-day hospital readmissions and (v) major cardiovascular-cerebrovascular events or deaths within 12 months of an episode of ischemic stroke [20].

### 1.2. Evaluation and Improvement of Quality in Post-Stroke Rehabilitation

Although the indicators calculated by the PNE for ischemic stroke allow for assessing the quality of care in acute wards, indicators to assess the quality of post-acute stroke rehabilitation are not available nationwide. Moreover, the literature shows limited studies on quality improvement strategies for stroke rehabilitation, although several studies have shown that monitoring rehabilitation outcomes could play an important role in improving the quality of care in these patients [23,24,25]. For this reason, the impact of quality interventions in improving stroke rehabilitation is still unclear [26,27].

Audit & Feedback (A&F) strategies can also be used to improve performance and promote best practices in post-stroke care [28,29], along with the use of quality indicators for outcome monitoring. Indeed, A&F is a set of interventions aimed at improving the quality of care through a systematic review of professional performance and monitoring of specific quality indicators. Although most studies concerning A&F strategies have focused on the area of emergency and acute stroke care, interest in evaluating guideline adherence in stroke care in rehabilitation has only recently grown [30,31].

### 1.3. The EASY-NET Research Program

The Italian network project EASY-NET—“Effectiveness of Audit & Feedback strategies to improve healthcare practice and equity in various clinical and organizational settings” is a multiregional research program funded by the Italian Ministry of Health, focused on evaluating various A&F strategies with the aim of improving quality of care in different health conditions and in different territorial and organizational contexts [32]. The network consists of seven Italian regions, each of which relates to a Work Package (WP) with a different scope within the program. As part of the EASY-NET program, the region of Sicily (WP7) is represented by the IRCCS Centro Neurolesi Bonino-Pulejo of Messina.

The WP7 designed a new study protocol aimed to assess, among other aspects, the effectiveness of A&F interventions in ischemic stroke settings, both in the acute and rehabilitation process of care [33]. To support this A&F methodology, WP7 designed a new set of rehabilitation QIs.

This paper describes the development of quality indicators to measure the performance in post-acute stroke rehabilitation, reporting the preliminary results obtained from their application on data collected in a group of Sicilian hospitals from 2019 to 2021.

## 2. Materials and Methods

This study employed a standardized approach to develop quality indicators for ischemic stroke rehabilitation based on a systematic literature review, a rating of published evidence, and an external peer review [26]. The study was performed from March 2022 to April 2023.

According to [34], three important issues have been considered: 1. the stakeholder perspective that indicators intended to reflect; 2. the aspects of health care measured; and 3. the evidence available. Thus, we chose to focus primarily on outcome QIs because they can be easily obtained through hospital administrative data, in addition to being the most effective tool to measure performances in a neurorehabilitation setting [35]. Figure 1 represents the whole process of QIs development applied in this study.

Initially, a systematic literature search of PubMed, Embase, and Web of Science databases from 2006 to 2021 was conducted, looking for existing rehabilitation QIs in post-ischemic stroke care. Adopted search terms were “stroke” and “rehabilitation” within the title field and the term “indicators” within the title or abstract fields. Publications that were not in English and did not correspond to the “Article” type were excluded from the search. On the basis of the literature review, we extract a list of QIs to submit to a panel of five stroke rehabilitation experts, also including new QIs defined from scratch.

The panel of experts included neurologists and healthcare quality specialists (e.g., physiatrists) with extensive experience in stroke rehabilitation, external to the EASY-NET project but employees of the IRCCS. The expert group used the consensus method to conduct the panel meetings, discussing and selecting the most suitable QIs to be part of the final set [36,37]. The process was carried out in two stages: in the first stage, each expert individually gave his/her assessment and thoughts on each of the proposed indicators, while in the second stage, the rankings were tabulated and presented during a meeting to discuss the points made by each expert.

Subsequently, a feasibility study was performed on the selected list of indicators, verifying their applicability and implementation based on available data sources, followed by a pilot test to assess the validity of the indicator set. For each indicator, we specified: the definition, the formula for calculating its value, and the type of indicator. The pilot test was conducted with the support of the Department of Health and the Sicily Region Epidemiological Observatory (DASOE), calculating annual indicators from 2019 to 2021 in four Sicilian neuro-rehabilitation facilities.

The final list of indicators was included in an A&F model under the EASY-NET project, with a view to providing a strategy intervention for improving the care services. Notably, the WP7 audit group discussed the developed QIs with the multidisciplinary rehabilitation team of any respective facility and feedback on their comments concerning the effectiveness of the established indicators in describing the quality of the rehabilitation outcome.

### Data Source and Study Population

Data were extracted from the regional SDO database, including information on all hospitalizations provided in public and private hospital facilities throughout Sicily. The information included patient demographic characteristics (e.g., age, sex, residence, education level), admission characteristics (e.g., hospital and discharge department, admission and discharge type, reservation date, admission priority class), and clinical characteristics (e.g., principal diagnosis, secondary diagnoses, diagnostic or therapeutic procedures). For the classification of diagnoses and diagnostic and therapeutic procedures, the SDO flow uses the ICD-9-CM coding (International classification of diseases, Clinical modification) [38].

The study population covered patients referred to four Sicilian neuro-rehabilitation facilities selected at the beginning of the EASY-NET project [32]: two in the city of Messina (NRF1 and NRF2), one in the city of Palermo (NRF3), and one in the city of Catania (NRF4). Notably, it includes all patients having hospital discharge records with a secondary diagnosis of ischemic stroke (ICD-9-CM codes 433.x1, 434.x1, 436) and a primary diagnosis within the following: late effects of cerebrovascular disease (ICD-9-CM codes 438.xx), hemiplegia (ICD-9-CM codes 342.xx), other paralytic syndromes (ICD-9-CM codes 344.xx), dysphagia (ICD-9-CM Code 787.2), abnormal involuntary movements or abnormality of gait (ICD-9-CM codes 781.0, 781.2). Patients younger than 34 and older than 100 years were excluded from the cohort. Furthermore, admissions lasting less than two days and discharged home were excluded, as well as hospitalizations of not Italian residents, admissions preceded by an acute discharge for ischemic stroke from more than 6 months, admissions preceded in the last 1 year by another rehabilitative hospitalization for ischemic stroke, and hospitalizations with a diagnosis of hemorrhagic stroke (ICD-9-CM codes 430, 431, 432.x).

## 3. Results

### 3.1. Review of the Literature and Qis Extraction

In the literature search, after duplicates removing, a total of ninety-seven search results were screened. Of these, only five articles described the use of QIs to measure process or outcome in stroke rehabilitation [26,27,39,40,41]. However, we found that only two out of five included the development of QIs. Grube et al. [26] reported the development of evidence-based quality indicators for stroke rehabilitation in Germany, according to a systematic literature review, rating of published evidence, an external peer review, and evaluation in a pilot study before implementation. These authors adopted a final set of eighteen indicators (nine processes, five outcomes, and four structure QIs). Miura et al. [39] used an evidence-based approach to develop fifteen indicators for stroke rehabilitation in Japan (five processes, eight outcomes, and two structure QIs) from identified reports and guidelines and survey educational hospitals certified by the Japanese Association of Rehabilitation Medicine, discussing them with an expert panel.

On the basis of information from these sources, we extracted only two candidate QIs: Home Discharges (HD), i.e., the rate of discharges to home after hospitalization for ischemic stroke rehabilitation; Average Length of Stay in Rehabilitation (ALoSR), i.e., the average days of hospitalization for ischemic stroke rehabilitation. However, we proposed an additional seven candidate QIs defined from scratch by the W7 team.

The list of nine candidate indicators (seven outcomes and two process QIs) is reported in Table 1.

The Number of Rehabilitation Admissions (NoRA) is a key indicator for monitoring and comparing the post-stroke rehabilitation outcomes, allowing the performance of an organization to be observed over time at a quantitative level. Long waiting times can affect the outcome of the rehabilitation process after an ischemic stroke event. Therefore it is essential to minimize the waiting time for admission. The Average Waiting Time for Rehabilitation (AWTR) indicator aims to assess the time in days from the discharge from acute wards to admission to rehabilitation care. Discharging the patient to his or her home is highly desirable. A patient discharged at home is an indication of a positive outcome of the rehabilitation pathway. Furthermore, post-stroke rehabilitation has been shown to positively affect the probability of the patient being discharged at home [42]. The HD indicator is used to evaluate this performance in a hospital facility. A protected discharge is the planned and agreed transition of a patient from a hospital to another care setting. It is applied in agreement with the patient and involves coordination between the general practitioner and health services in the patient’s home area. Protected discharges can occur in various forms depending on the services active in the patient’s home area and his clinical condition. After a protected discharge, the patient can return to his home by activating a home care pathway or being housed in a nursing home. The Protected Discharges indicator (PD) shows the proportion of this type of discharge. Patients with stroke who are readmitted to acute care during the rehabilitation process suffer an interruption in their individualized plan of care, affecting the progression of their treatment plan [43]. For this reason, reducing acute readmissions during rehabilitation is a critical goal. The Acute Discharges (AD) indicator provides an assessment of the proportion of hospitalizations interrupted by readmission to acute care. Comparing the outcomes of HD, PD, and AD indicators can serve as a valuable benchmark for assessing the performance of the post-acute stroke rehabilitation pathways. The ALoSR indicator shows the average number of days that patients spend in rehabilitation hospitalizations. The Medical Complications in Rehabilitation (MCR) indicator provides the proportion of rehabilitation admissions with medical complications such as falls, healthcare-associated infections, and acute state recurrence. A link may exist between the length of stay and the incidence of complications during hospitalization, although it is difficult to determine whether one causes the other [44]. The Average Change in Level of Disability (ACLoD) indicator measures the changes in the patients’ levels of disability from admission to discharge during the rehabilitation programs. The level of disability defined as the independence in activities of daily living is assessed using the original Barthel index, which is scored 0–100 points, with higher scores indicating higher daily independence [45]. For each rehabilitation program, based on the patient’s condition, a goal to be achieved is assigned at the beginning of the process. Goals are set by both the rehabilitation care team and the attending physician. In general, the goals are related to the achievement of autonomy in the activities of daily living and the regaining of skills that condition personal social and economic disadvantage. A goal may or may not be achieved at the time of discharge. The Rehabilitation Goals Achievement (RGA) indicator is the proportion of admissions in which rehabilitation goals were achieved.

### 3.2. Expert Panel Approval and Final List of QIs

The expert panel approved seven of the nine proposed indicators, with an overall positive response. The two excluded indicators were: MCR and ALoSR because they did not believe that the average correctly represented the size of the length of stay for this (often highly variable) type of hospitalization.

The expert panel discussed the lack of QIs measuring the recovery due to neuro-rehabilitation, such as physical therapy for patients with gait disturbance and occupational therapy for patients affected by mild upper limb dysfunctions. However, these items were eventually dismissed based on measurement uncertainty.

### 3.3. Feasibility and Pilot Study

The remaining seven QIs were mathematically defined by checking their applicability and implementation based on available data sources; the formulas are reported in Appendix A. Therefore, a pilot study of the feasibility of the indicators was conducted.

Based on the results of the feasibility study, it was decided to discard two QIs: ACLoD and RGA, because their calculation required information contained in the medical records and in the regional SDO database. Table 2 shows the results of the pilot test for the five QIs calculated.

The NoRA indicator shows a decrease in the number of rehabilitation admissions over the three years (Figure 2).

Results of the other QIs are shown in Figure 3.

A standardized audit was conducted by using these five QIs for any rehabilitation facility included. Thus, a report form on center performance was provided and discussed. Feedback from the participating centers on the feasibility and content of the report form was mainly positive, and the QIs were comprehensible and appreciated.

## 4. Discussion

There are numerous studies in the literature outlining methodologies for measuring stroke care quality in the acute setting, as suggested by a recent review on the topic [46]. On the contrary, the literature highlighted the lack of studies focused on post-stroke rehabilitation indicators [36]. However, rehabilitation indicators may be employed to determine whether disparities exist in providing rehabilitation care services [35].

To the best of our knowledge, this is the first study focused on improving the quality of post-stroke rehabilitation care services in Italy. We developed a set of quality indicators from previous research and predefined methodological requirements. The indicators were discussed by an expert panel in stroke rehabilitation. The feedback received from the panel of experts was overall positive. Involving an expert panel was an important step in detecting any critical issues in the proposed indicators. The set of QIs has been tested in a retrospective pilot study that demonstrated their feasibility before being implemented in a group of participating centers. Finally, this set of QIs has been used to conduct an A&F intervention.

In accordance with the recommendations established in the national and international guidelines, all hospitalized patients with stroke should be able to take advantage of specialized rehabilitation services equipped to meet their health, social, and work/education needs as soon as possible after the acute discharge. Indeed, long waiting times for admission to the rehabilitation unit can affect the outcome of the treatment after an ischemic stroke event [47]. In addition, after the neuro-rehabilitation treatment, an early supported discharge service to facilitate rehabilitation at home or at a residential center for all patients with stroke (i.e., protected discharge) should also be provided [48]. For this reason, we considered indicators such as AWTR, HD, and PD. Indicators describing the discharge destination can be valid tools to assess the overall quality of rehabilitation pathways, especially when evaluated in combination. Increasing the number of discharges to the patient’s home, along with decreasing acute transfers and medical complications during rehabilitation hospitalization, are the goals identified by our study for achieving quality improvement.

The set of process and outcome indicators used in this study, relevant only to inpatient settings, can be a valuable starting point for enabling specialists to monitor the functioning of stroke rehabilitation units. The results of the audit showed that these indicators identify some strengths and areas for improvement in stroke management. Larger studies could confirm their usefulness in reorganizing healthcare services and procedures, as proposed by other studies [49].

This study has several limitations. First, to avoid refusal to participate in the study, we included rehabilitation facilities in Sicily only, all belonging to the same neuro-rehabilitation network [50], and this substantially reduced the cohort of patients. One might think that the results obtained are those of elite facilities because the selected hospitals all have one or more certified physiatrists, leading to better clinical management and functional improvement in neurological rehabilitation [51]. However, by expanding the cohort and analyzing more hospitals in a wider geographic area, we expect to see more consistent results. Second, the indicators could be difficult to adapt to other healthcare systems. In Italy, rehabilitation and acute care are offered in separate units, but it could be different in other countries. In addition, our indicators did not evaluate nonmedical aspects of care, including neuro-psychology, speech therapy, physiotherapy, social work, and nursing care. Indeed, we chose to implement only those QIs that did not require inspection of medical records but rather hospital administrative data, which are easily accessible and cost-effective, in addition to covering large populations over long periods. Moreover, this type of data makes it possible to monitor and report recovery without being affected by the subjectivity of the rehabilitation treatment providers. Therefore, administrative data are useful screening tools to identify areas in which quality should be investigated more thoroughly through audits. Despite these beneficial characteristics, it is also important to consider the limitations that administrative health research presents, such as issues related to data accuracy and incompleteness [52]. Indeed, there is a need for interventions that instill staff with greater care in compiling health administrative data. Furthermore, administrative data do not provide quantification of the “dose” of rehabilitation received after stroke, which is important to patient outcomes after stroke [53,54,55]. Third, there are no databases or platforms to use to compare and verify QIs. Lastly, results were severely affected by the COVID-19 pandemic. The pandemic has had an impact on both admission (significantly decreased, even in scheduling) and modalities of patient discharges. For these reasons, performance measurements should be carried out over the next years in order to see results not influenced by COVID-19. However, the appreciation of the healthcare staff being audited is encouraging, although the study highlighted the need for better adherence to indicators measuring processes of rehabilitation care.

Other areas that could be considered in future research include engaging in knowledge sharing, joint research projects, and collaborative initiatives to enhance post-stroke care delivery, promote best practices, and support the development of stroke care networks.

## 5. Conclusions

In Italy, the absence of national quality indicators models for post-acute rehabilitation of stroke patients represents a major limitation in measuring the quality of stroke treatment. In our view, more consideration should be given to quality measurement in post-stroke rehabilitation. This study may contribute to national policy research and countermeasures for quality improvement in medical research.

## Figures and Tables

**Figure 1 healthcare-11-02065-f001:**
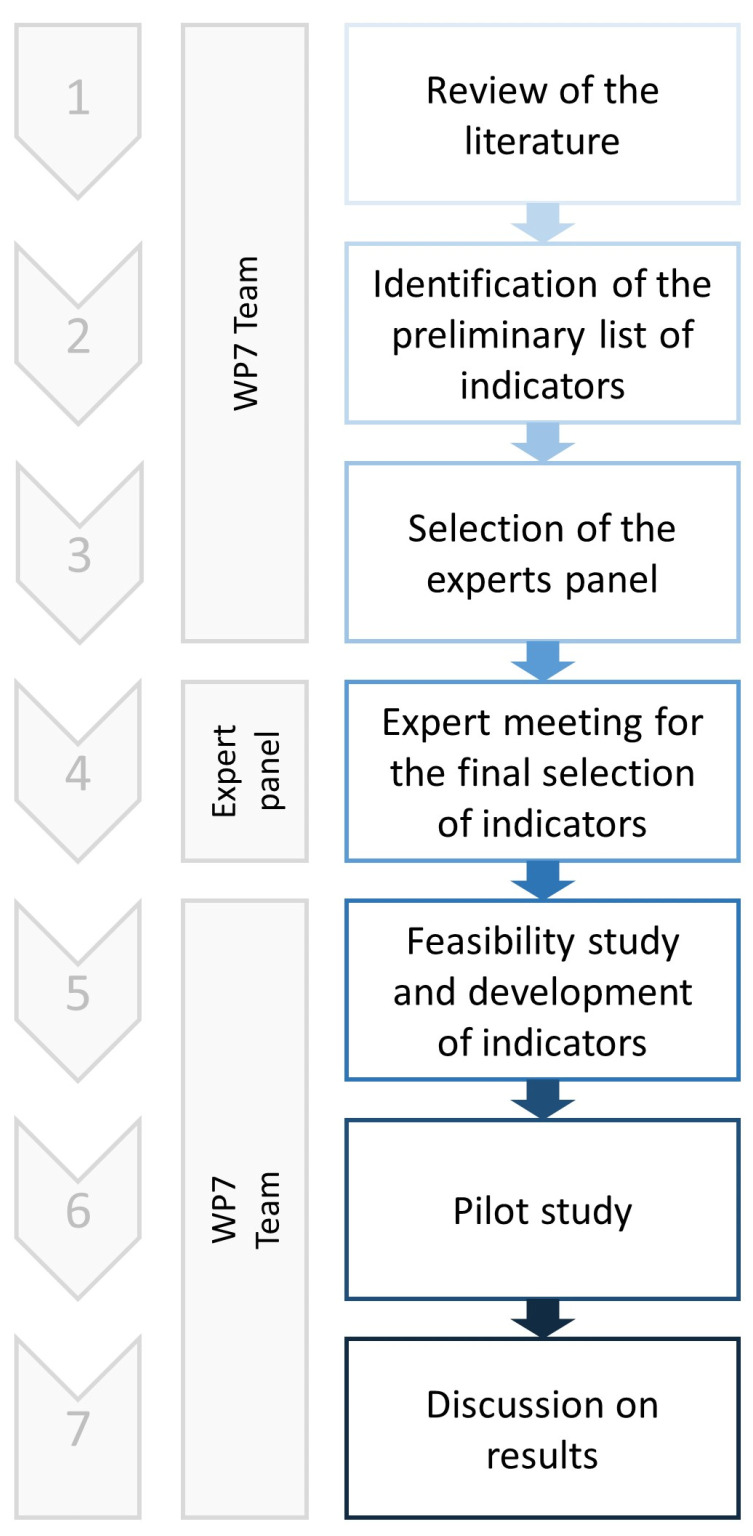
Flow of the study. Legend: WP7 = Work Package 7.

**Figure 2 healthcare-11-02065-f002:**
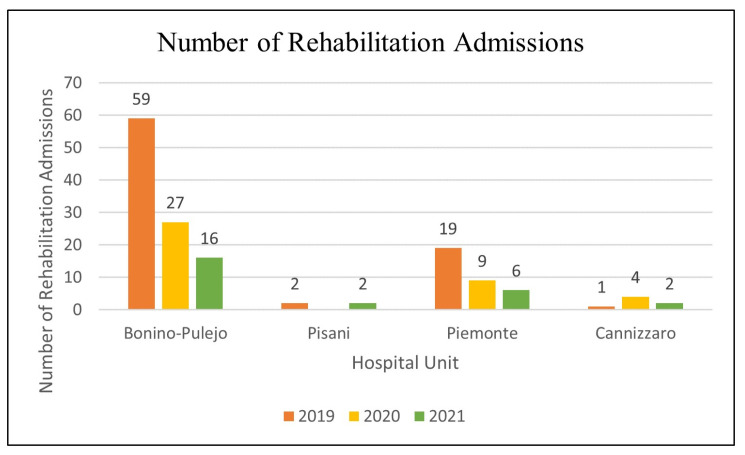
Results of pilot testing: Number of Rehabilitation Admissions.

**Figure 3 healthcare-11-02065-f003:**
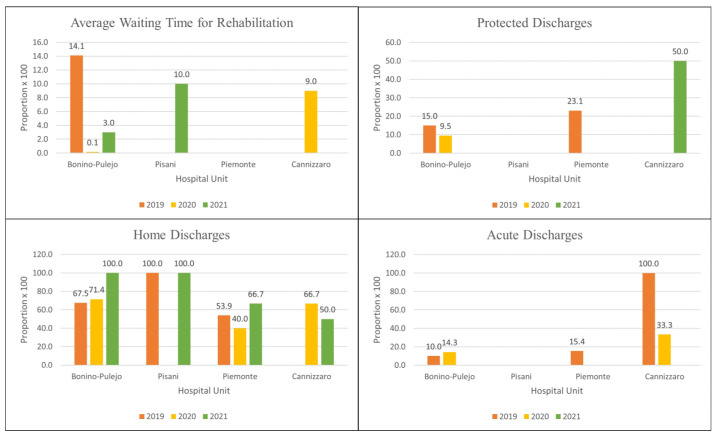
Results of pilot testing: Results of AWTR, HD, PD, and AD indicators. Legend: AWTR = Average Waiting Time for Rehabilitation, HD = Home Discharges, PD = Protected Discharges, AD = Acute Discharges.

**Table 1 healthcare-11-02065-t001:** The initial list of indicators submitted to the expert panel.

Indicator	Description	Type
NoRA (Number of Rehabilitation Admissions)	Number of rehabilitation admissions for ischemic stroke	Outcome
AWTR (Average Waiting Time for Rehabilitation)	Average waiting time for rehabilitation admission following the discharge from acute hospitalization for ischemic stroke	Process
HD (Home Discharges)	Proportion of discharges to the patient’s home following the admission to rehabilitation for ischemic stroke	Outcome
PD (Protected Discharges)	Proportion of protected discharges following the admission to rehabilitation for ischemic stroke	Outcome
AD (Acute Discharges)	Proportion of transfers to acute care following the admission to rehabilitation for ischemic stroke	Outcome
ALoSR (Average Length of Stay in Rehabilitation)	The average number of days that patients spend in rehabilitation	Process
MCR (Medical Complications in Rehabilitation)	Proportion of rehabilitation admissions with medical complications	Outcome
ACLoD (Average Change in Level of Disability)	Average change in the level of disability from admission to discharge	Outcome
RGA (Rehabilitation Goals Achievement)	Proportion of admissions in which rehabilitation goals were achieved	Outcome

Legend: Indicator = Name of the indicator, Description = Short description of the indicator, Type = Type of the indicator (Process or Outcome).

**Table 2 healthcare-11-02065-t002:** Results of pilot testing.

	2019					2020					2021				
** Hospital Unit **	NoRa	AWTR	HD	PD	AD	NoRa	AWTR	HD	PD	AD	NoRa	AWTR	HD	PD	AD
**Bonino Pulejo**	59	14.10	67.50	15.00	10.00	27	0.14	71.43	9.52	14.29	16	3.00	100.00	0.00	0.00
**Pisani**	2	0.00	100.00	0.00	0.00	0	0.00	0.00	0.00	0.00	2	10.00	100.00	0.00	0.00
**Piemonte**	19	0.00	53.85	23.08	15.38	9	0.00	40.00	0.00	0.00	6	0.00	66.67	0.00	0.00
**Cannizzaro**	1	0.00	0.00	0.00	100.00	4	9.00	66.67	0.00	33.33	2	0.00	50.00	50.00	0.00

Legend: Mean was used to describe continuous variables; proportions were used to describe categorical variables. NoRA = Number of Rehabilitation Admissions, AWTR = Average Waiting Time for Rehabilitation, HD = Home Discharges, PD = Protected Discharges, AD = Acute Discharges.

## Data Availability

The aggregated data analyzed in this study were obtained from the Department of Health and Epidemiological Observatory of Sicily Region (DASOE). Datasets are available in a publicly accessible repository under request.

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
