# Peer review of "Development of a Set of Indicators for Measuring and Improving Quality of Rehabilitation Care after Ischemic Stroke"

_healthcare, 2023, doi:10.3390/healthcare11142065_

Round 1

Reviewer 1 Report

Thank you for your work. This paper examined the development and implementation of quality indicators for stroke rehabilitation.

The work is sound and explains the clear gap in literature and how the authors intend to address this.

Check references section- references are numbered twice.

In some paragraphs, there was switching between present and past tense which disrupted the flow of reading. Minor changes can be made to address this. e.g. discussion line 27. 

Author Response

Dear Reviewer,

we are grateful for your insightful comments on our paper. Please find our answers below.

"Check references section- references are numbered twice."

Author: Thank you for pointing out the issue. References have been revised and corrected.

Reviewer 2 Report

This is an interesting study reporting the development of a set of indicators to measure and improve quality of rehabilitation care after ischemic stroke. I have some suggestions.

ABSTRACT: please spell out all the acronyms the first time you cite them (Line 2 QI is not spelled out)

Line 23-24 I think the English form of the sentence may be improved. I suggest to change ‘ it is emerged the need of a better adherence…’ with this sentence: ‘ The study outlined (or highlighted) the need of a better adherence…’

INTRODUCTION:

-          LINE 32: change ‘is’ with ‘are’ (the subject is plural)

-          1.1 Measuring stroke care perfomances, line 53-54 I think the English form of the sentence may be improved. I suggest to change ‘ Constant review of clinical outcomes establishes standards against which to continuously improve healthcare practice’ with this sentence: ‘Constant review of clinical outcomes establishes standards which may be addressed to promote continuous improvement in healthcare practice’

-          LINE 62-63 ‘In Italy, several research programs on outcomes have been conducted during the last 20 years’, please give reference

-          LINE 79 please give reference

METHODS

2.1 Data source and study program

- Line 155, Information included patient demographic characteristics… (I suggest to maintain the past form of verbs)

RESULTS

-          Line 233: When you refer to Barthel Index please check the reference: different versions exist, so you have to explain if you refer to the original BI or to the Modified, etc.

-          Table 2: I suggest to separate data regarding the different years, in this form the Table is difficult to be read; even a simple bold vertical line could enhance the distinction of different years.

DISCUSSION:

- line 273-274: authors say ‘There are numerous syudies’, but they report only one reference. Please add references.

- lines 292-309: I think there is a typo in reference number 44-47, please check.

-line 300: why do you say the study identified process and outcome indicators to monitor functioning of stroke units? You focused on rehabilitation care till now

-lines 330-331 like in the Abstract, I think the English form of the sentence may be improved. I suggest to change ‘ it is emerged the need of a better adherence…’ with this sentence: ‘ The study outlined (or highlighted) the need of a better adherence…’

- moreover I suggest to enrich the discussion in the part regarding the expert panel feedbacks. This could be useful for other researchers.

- I suggest to include something about other countries, for example which quality indicators are used to monitor healthcare stroke pathways in other European countries or USA, etc. This could help researchers to focus on feasible process or outcome indicators to be used in their context.

- When I made the download of supplementary material and non-published materials, I found the same file, did you want to upload something different?

Some sentences may be improved (see previous comments)

Author Response

Dear Reviewer,

we are grateful for your insightful comments on our paper. Please find our answers below.

ABSTRACT

"Please spell out all the acronyms the first time you cite them (Line 2 QI is not spelled out)."

Author: Done.

"Line 23-24 I think the English form of the sentence may be improved. I suggest to change ‘ it is emerged the need of a better adherence…’ with this sentence: ‘ The study outlined (or highlighted) the need of a better adherence…’"

Author: Thank you for your suggestion. The sentence was changed.

INTRODUCTION

"LINE 32: change ‘is’ with ‘are’ (the subject is plural)"

Author: Done.

"1.1 Measuring stroke care perfomances, line 53-54 I think the English form of the sentence may be improved. I suggest to change ‘ Constant review of clinical outcomes establishes standards against which to continuously improve healthcare practice’ with this sentence: ‘Constant review of clinical outcomes establishes standards which may be addressed to promote continuous improvement in healthcare practice’"

Author: Thank you for your suggestion. The sentence was changed.

"LINE 62-63 ‘In Italy, several research programs on outcomes have been conducted during the last 20 years’, please give reference"

Author: The research programs names are now cited in the text. The related reference item is #19.

"LINE 79 please give reference"

Author: Item reference #20 was moved at the end to give reference to the entire paragraph.

METHODS

"2.1 Data source and study program

- Line 155, Information included patient demographic characteristics… (I suggest to maintain the past form of verbs)"

Author: Done.

RESULTS

"- Line 233: When you refer to Barthel Index please check the reference: different versions exist, so you have to explain if you refer to the original BI or to the Modified, etc."

Author: We confirm that the original BI was used. This information was added to the manuscript as suggested.

"Table 2: I suggest to separate data regarding the different years, in this form the Table is difficult to be read; even a simple bold vertical line could enhance the distinction of different years."

Author: Done.

DISCUSSION

"- line 273-274: authors say ‘There are numerous syudies’, but they report only one reference. Please add references."

Author: Thank you for your suggestion. The reported reference is a recent review on the topic. We added this information to the manuscript.

"- lines 292-309: I think there is a typo in reference number 44-47, please check."

Author: Thank you for pointing out the issue. Reference numbers were revised and corrected.

"-line 300: why do you say the study identified process and outcome indicators to monitor functioning of stroke units? You focused on rehabilitation care till now"

Author: The word rehabilitation is indeed missing in the sentence. Manuscript was updated. Thank you for pointing it out.

"-lines 330-331 like in the Abstract, I think the English form of the sentence may be improved. I suggest to change ‘ it is emerged the need of a better adherence…’ with this sentence: ‘ The study outlined (or highlighted) the need of a better adherence…’"

Author: Done.

"- moreover I suggest to enrich the discussion in the part regarding the expert panel feedbacks. This could be useful for other researchers."

Author: Thank you for your suggestion. The part regarding the expert panel feedbacks was enriched.

"- I suggest to include something about other countries, for example which quality indicators are used to monitor healthcare stroke pathways in other European countries or USA, etc. This could help researchers to focus on feasible process or outcome indicators to be used in their context."

Author: Thank you for your suggestion. Our study is focused on stroke rehabilitation, and the search we did in the literature is about developing new quality indicators for stroke rehabilitation. The results obtained from our literature review are too limited to give an indicative overview. However, we have added the country reference in the results section (3.1. Review of the literature and Qis extraction).

"- When I made the download of supplementary material and non-published materials, I found the same file, did you want to upload something different?"

Author: Thank you for pointing that out. We confirm that we did not want to upload a different file.

Reviewer 3 Report

“Development of a set of indicators for measuring and improving quality of rehabilitation care after ischemic stroke.”

Overall strengths of the article:

This study describes the development of quality indicators to measure the performance in post-acute stroke rehabilitation, reporting the preliminary results obtained from their application on data collected in a group of pilot tests conducted on a small sample of Sicilian rehabilitation facilities from 2019 to 2021. Feedback from the participating centers was mainly positive and the quality indicators were found to be comprehensible and appreciated. However, it has emerged the need for better adherence to indicators measuring processes of rehabilitation care. Implementing effective strategies for preventing and treating stroke plays a key role in clinical risk management. The set of quality indicators presented in this study could be considered a starting point on which to base quality improvement initiatives both nationally and internationally. Overall, the methods are well-detailed, and the data is well-presented (except for maybe a few critiques detailed below).

Specific comments on weaknesses:

Major Critical Comments:

1.      Improving performance and promoting best practices in post-stroke care involves a comprehensive approach that addresses various aspects of patient care: why key performance indicators, clinical outcomes, and patient satisfaction were not included in this study?

2.      Sample size is too small to make any definitive conclusion.

3.      Engaging in knowledge sharing, joint research projects, and collaborative initiatives to enhance post-stroke care delivery, promote best practices, and support the development of stroke care networks could be the other areas that should also be carefully considered.

Minor points:

1.      References need to be formatted carefully each citation is numbered twice in the present manuscript.

moderate English correction and formatting needed throughout the whole manuscript.

Author Response

Dear Reviewer,

we are grateful for your insightful comments on our paper. Please find our answers below.

Major Critical Comments:

"1. Improving performance and promoting best practices in post-stroke care involves a comprehensive approach that addresses various aspects of patient care: why key performance indicators, clinical outcomes, and patient satisfaction were not included in this study?"

Author: Thank you for your insights on performance improvment and promotion of best practices in post-stroke care. We agree with you on the importance of process indicators and clinical outcomes. However, as we state in our paper in the study limitations section, we chose to implement only those QIs that did not require inspection of medical records, but rather hospital administrative data, which are easily accessible and cost-effective.

"2. Sample size is too small to make any definitive conclusion."

Author: You are right about the sample size. We can consider this as a pilot study. However, from the results of the audit, it seems that the proposed quality indicators have been well received by the participating facilities.

"3. Engaging in knowledge sharing, joint research projects, and collaborative initiatives to enhance post-stroke care delivery, promote best practices, and support the development of stroke care networks could be the other areas that should also be carefully considered."

Author: Thank you for your feedback. Your suggestions have been added to the end of the discussion section as proposals for future research.

Minor points:

"1. References need to be formatted carefully each citation is numbered twice in the present manuscript."

Author: Thank you for pointing out the issue. References have been revised and corrected.

Round 2

Reviewer 2 Report

I thank all the authors for the responses point-by-point. I think now the paper has been improved and it is clear and interesting.

Author Response

Dear Reviewer, thank you for your feedback.